# An Investigation of Cutting Performance and Action Mechanism in Ultrasonic Vibration-Assisted Milling of Ti6Al4V Using a PCD Tool

**DOI:** 10.3390/mi12111319

**Published:** 2021-10-28

**Authors:** Yongsheng Su, Liang Li

**Affiliations:** 1School of Mechanical Engineering, Anhui Polytechnic University, Wuhu 241000, China; 2College of Mechanical and Electrical Engineering, Nanjing University of Aeronautics & Astronautics, Nanjing 210016, China; liliang@nuaa.edu.cn

**Keywords:** Ti6Al4V titanium alloy, UVAM, cutting force, surface roughness, action mechanism

## Abstract

A polycrystalline diamond (PCD) tool is employed in cutting various titanium alloys because of its excellent properties. However, improving the cutting performance of titanium alloys is still a challenge. Here, an experimental investigation on the influence of ultrasonic vibration-assisted machining (UVAM) of Ti6Al4V titanium alloy on the cutting performance and action mechanism was studied using a PCD tool. Cutting force, machined surface, surface adhesion, and wear morphology were analyzed. The results indicated that UVAM can effectively improve cutting performance. It was found that there was serious adhesion and wear of slight fragments close to the cutting edge after ultrasonic-assisted dry milling. Furthermore, the action mechanism of UVAM in improving cutting performance was discussed and analyzed from the perspective of intermittent cutting.

## 1. Introduction

Currently, various types of titanium alloys have been increasingly used in aircraft, rockets, missiles, automobiles, ships, and biological materials; titanium alloys occupy an extremely important position because of their excellent properties [1,2,3]. However, the poor machinability of titanium alloys restricts their application to a large extent and results in lower machining efficiency and shorter tool life as compared with other conventional materials [4,5,6]. Therefore, it is of great significance to obtain high efficiency machining performance of titanium alloys and expand their application field.

Various methods to improve a titanium alloy’s machinability include employing surface coating [7], a superhard tool [8], cryogenic lubrication [9], structure optimization of the tool [10], bionic surface texture [11], and micromachining processes [12,13,14], etc. In recent years, ultrasonic machining methods have been widely studied for improving cutting performance at home and abroad. Gao et al. [15] found that ultrasonic-assisted milling using a new cutting force model had great effects on reducing the cutting force as compared with conventional milling. Chen et al. [16] found that the method of ultrasonic machining of composites effectively decreased cutting temperature and improved machining quality. Ni et al. [17] studied the performance of TC4 alloy using a carbide cutter and the results indicated that the ultrasonic machining had a better effect in reducing surface roughness and cutting force than that of the common milling method under the same condition. Ni et al. [18] also found that better surface quality could be obtained using the UVAM method with or without MQL as compared with that of conventional milling (CM), which can contribute to reducing tool wear. Lu et al. [19] studied the influence of UVAM on titanium alloy and the results demonstrated that a combination of the high-speed ultrasonic vibration and high-pressure coolant had a positive effect on the cutting performance. It is well known that a polycrystalline diamond (PCD) tool has the characteristics of super hardness and excellent wear resistance.

Some studies have reported that a PCD tool has superior properties of anti-friction, anti-wear, and anti-adhesion as compared with other common machining tools [20,21,22,23,24]. In order to reduce environmental pollution caused by using cutting fluid, dry cutting has been increasingly used in machining of titanium alloys, which can effectively reduce machining costs and avoid the environmental pollution from chip dust and harm to workshop personnel [25]. However, few studies have reported on the machinability of ultrasonic-assisted dry milling of titanium alloy using a PCD tool. In this study, we conduct a comparative investigation of the effect of two methods, i.e., CM and UVAM, on the cutting performance and action mechanism using a PCD tool in dry cutting.

## 2. Experimental Methods and Materials

The slot milling experiments of CM and UVAM are carried out on a milling machine under different machining parameters. In addition, the relevant measurement equipment and instruments used in the experiment include a dynamometer (Kistler, type 9257B, Winterthur, Switzerland), scanning electron microscope (Phenom X, Eindhoven, The Netherlands), and a Mahr surface roughometer (Perthometer M1, Mahr, Germany). The cutting test apparatus are shown in Figure 1.

The milling machine, workpiece, structure parameters of the tool, rake angle, clearance angle, cutting parameters, lubrication method, and parameters of UVAM used in the study are listed in Table 1.

## 3. Results and Discussion

### 3.1. The Influence of UVAM on Cutting Force

In this part of the study, the values of selected a_p__,_ f_z_ and n in Figure 2 were 0.3 mm, (0.03~0.08) mm/z and 6000 rpm, respectively. Similarly, the values of selected a_p__,_ f_z_ and n in Figure 3 were (0.1~0.5) mm, 0.03 mm/z and 6000 rpm, respectively.

Figure 2 illustrates the influence of f_z_ on the cutting force under the conditions of CM and UVAM, and the cutting forces of Fx, Fy, and Fz show a gradual trend with an increase in f_z_ ranging from 0.01 to 0.08 mm/z. Similarly, whether the condition of CM or UVAM, the results in Figure 3 indicate that the cutting force of three directions also increases with an increase in a_p_. Some studies have reported that a higher cutting parameter leads to a higher milling force [26,27], which can be attributed to increasing the sectional area of cutting layer.

According to the experimental results shown in Figure 2 and Figure 3, it can be seen that UVAM can effectively decrease the cutting force as compared with that of CM at the same cutting parameters. Moreover, as compared with CM, when the f_z_ ranged from 0.01 to 0.08 mm/z, the F_X_, F_Y_ and F_Z_ of UVAM were reduced by 0.8~4.1%, 0.3~2.8%, and 3~42%, respectively. As compared with CM, when the milling depth ranged from 0.1 to 0.5 mm/z, the F_X_, F_Y_ and F_Z_ of UVAM were reduced by 5.3~27%, 3.8~41.2%, and 29.5~65%, respectively.

These above results indicate that the UVAM method has a better cutting effect by reducing the cutting force as compared with that of the CM, which can contribute to reducing cutting friction and improving the cutting performance. The reason for the lower cutting friction using the UVAM method could be explained from the perspective of a reduction in its intermittent cutting [28].

### 3.2. The Influence of UVAM on Surface Quality

Surface roughness was measured and analyzed after CM and UVAM. The value of surface roughness was measured three times. In this part of the experiment, several groups of cutting parameters were listed in Table 2.

Figure 4 demonstrates the comparison results of surface roughness under the conditions of CM and UVAM; The measured values show that the Ra of UVAM is lower at the same cutting parameter ranging from No. 1 to No. 6. Moreover, as compared with CM at the same cutting parameter ranging from No. 1 to No. 6, the surface roughness of UVAM was reduced by 13.63%, 25.41%, 33.08%, 19.10%, 10.82%, and 20%, respectively.

The improvement of surface roughness using UVAM may be explained from the perspective of the real friction time, which is shorter than the separation time between them, which can contribute to reducing cutting friction and improving the quality of machined surfaces to a certain degree.

The partial SEM topography of Figure 4 is illustrated in Figure 5. The images in Figure 5a,c,e were obtained under the condition of CM, and the SEM images in Figure 5b,d,f were obtained under the condition of UVAM. In addition, Figure 5a,b employs the same cutting parameter of No. 1, the cutting parameter of Figure 5c,d is No.4, and the cutting parameter of Figure 5e,f is No. 6. It can be observed in Figure 5a,c,e that the machined surface has many scratches under the condition of CM, whereas, as can be seen in Figure 5b,d,f, the machined surface is smoother under the condition of UVAM. The above results indicate that a better machined surface and surface quality can be gained by UVAM as compared with that of CM at the same cutting parameter.

### 3.3. Surface Adhesion of UVAM

It can be observed from the PCD tool that there is a layer of adhesion material after the ultrasonic-assisted dry milling of the titanium alloy. Figure 6 and Figure 7 show the energy dispersive X-ray spectroscopy (EDS) map and results of the adhesion material on the surface of the PCD tool. The results indicate that the main elements of the EDS map of Figure 6a–f include titanium, carbon, aluminum, tungsten, and cobalt. It can be concluded that the carbon, tungsten, and cobalt came from the PCD tool, and the titanium and the aluminum came from the titanium alloy. In order to further analyze the surface adhesion material, shown in Figure 6g, the analysis results of the EDS in the selected circle area M are shown in Figure 6h. On the basis of the above results, we can confirm that the adhesive materials of Figure 6g came from the workpiece.

According to the analysis results of Figure 7a–g, we can confirm that the adhesive material on the flank face also came from the titanium alloy workpiece. Therefore, according to the results of Figure 6 and Figure 7, it can be found that there is still a lot of chip adhesion on the surface even under the condition of UVAM.

### 3.4. Tool Wear of UVAM

In this study, all ultrasonic cutting tests were performed by one PCD tool, and the anti-wear of PCD tool was investigated after finishing all the ultrasonic cutting tests. The chip adhesion materials of the PCD tool were corroded for a better effect of obsevation. The SEM images of wear after ultrasonic-assisted dry milling are shown in Figure 8. In this study, the cutting length is about 612 mm. It can be observed that there are tiny fragment and chips on the flank face and that there is almost no wear on the rake face, which may be attributed to the excellent anti-wear of the PCD tool, the shorter cutting length, and the UVAM with characteristics of intermittent cutting, etc. Therefore, the PCD tool shows good anti-wear performance in this cutting conditions.

### 3.5. Discussion on Effect of UVAM

The above experimental results demonstrated that UVAM has a significant role in improving cutting performance and there are some main perspectives on its action mechanism.

As compared with that of CM, UVAM has less actual cutting time, which can be attributed to its intermittent cutting. Therefore, UVAM has a much longer time of complete separation time between tool-workpiece and tool-chip, and thereby produces lower cutting force and better machined surface [29]. Some studies have reported that the action mechanism of UVAM in reducing cutting force could be mainly attributed to intermittent cutting, and the results have demonstrated that the cutting force of PCD tool and surface roughness of machined surface using UVAM could be decreased by up to 40% and 30%, respectively [17].

In addition, UVAM with characteristics of intermittent cutting can improve the machined surface [30] mainly because of two aspects. Firstly, the less cutting force reduces the cutting friction on the interface of the tool-workpiece, and thereby decreases the damage to the machined surface. Secondly, the characteristics of intermittent cutting in UVAM has a significant effect of chip breaking, which can decrease the chip entanglement, and thus improve the machined surface.

## 4. Conclusions

This experiments investigated the machinability difference of titanium alloys using a PCD tool under CM and UVAM. Several conclusions are summarized from the study:Compared with the CM method, the UVAM method can effectively decrease the cutting force even under dry milling, and when the selected value of f_z_ and a_p_ ranged from low to high, respectively, the corresponding cutting force of UVAM was reduced by up to 0.8–42% and 5.3–65%, respectively.Compared to CM, UVAM can effectively reduce surface roughness, Ra, and the surface roughness can be reduced by 10.82–37.97% under the cutting condition of this study. The results demonstrate that UVAM is superior to CM in improving surface quality even under the dry milling condition.Under the condition of UVAM, the experimental results indicate that there is serious adhesion on the PCD tool, and there is only wear of tiny fragment on its flank face.The machining mechanism of UVAM can be mainly explained from the perspective of a reduction in actual contact time and an increase in separation time between the tool-workpiece and tool-chip owing to ultrasonic vibration with the characteristic of intermittent cutting.

## Figures and Tables

**Figure 1 micromachines-12-01319-f001:**
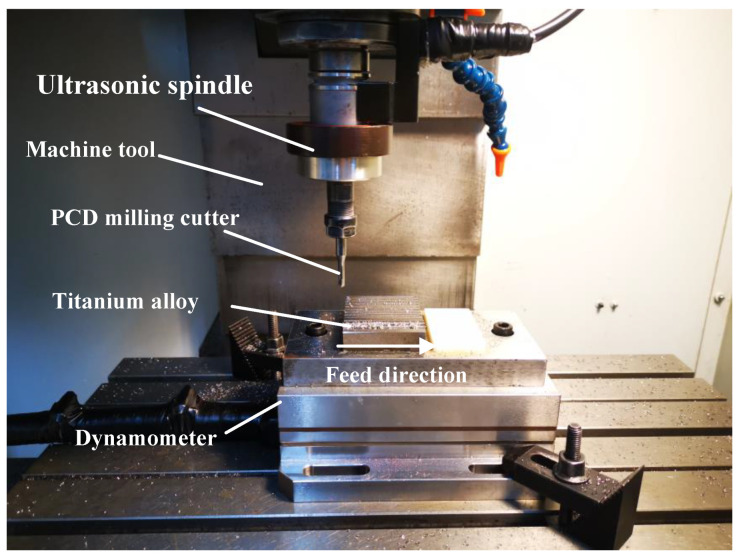
Cutting test apparatus.

**Figure 2 micromachines-12-01319-f002:**
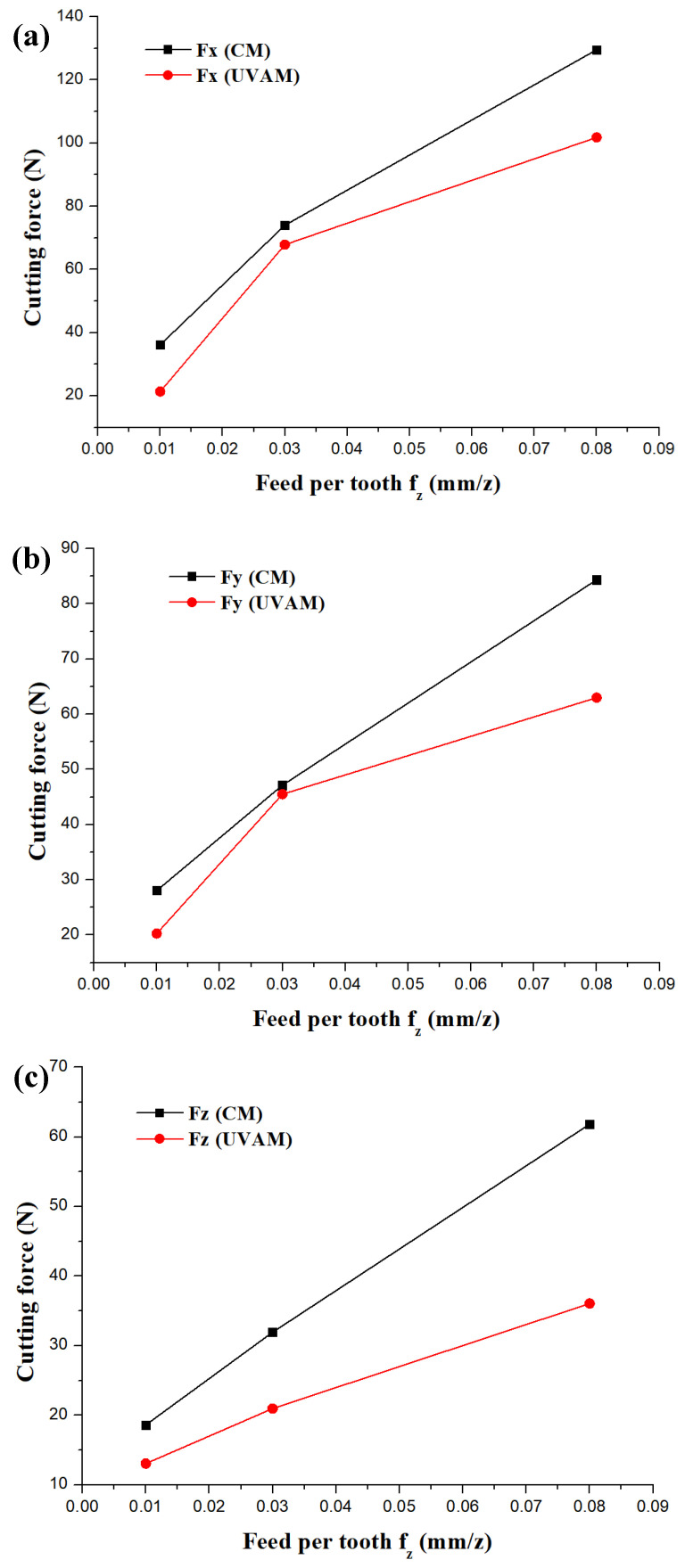
Influence of f_z_ on the cutting force of CM and UVAM. (**a**) changing trend of Fx, (**b**) changing trend of Fy, (**c**) changing trend of Fz.

**Figure 3 micromachines-12-01319-f003:**
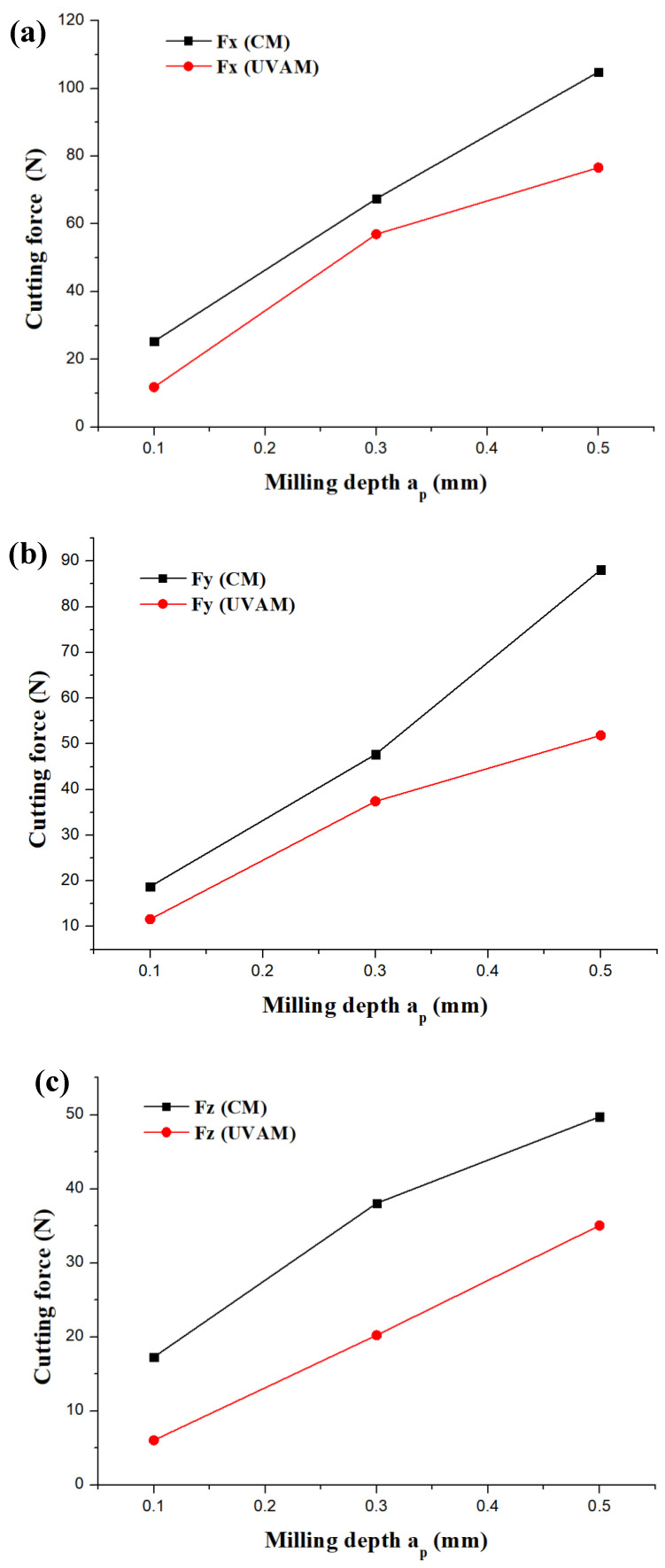
Influence of a_p_ on the cutting force of CM and UVAM. (**a**) changing trend of Fx, (**b**) changing trend of Fy, (**c**) changing trend of Fz.

**Figure 4 micromachines-12-01319-f004:**
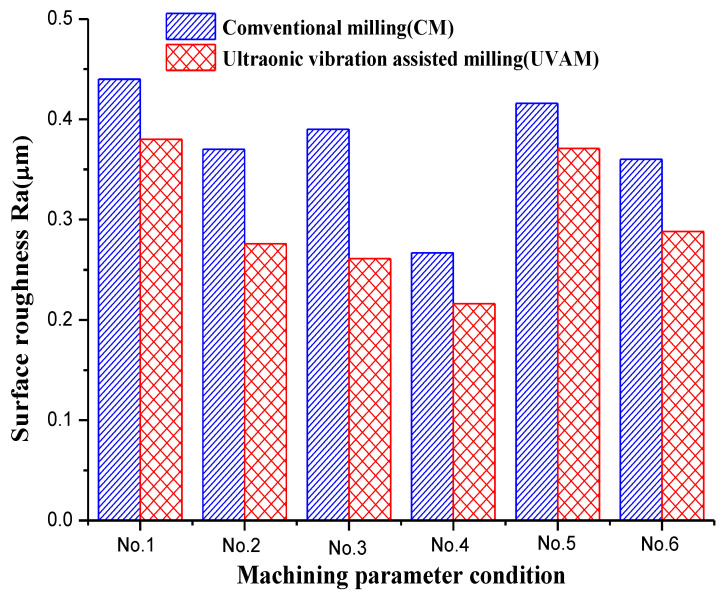
Comparison of surface roughness under different parameters.

**Figure 5 micromachines-12-01319-f005:**
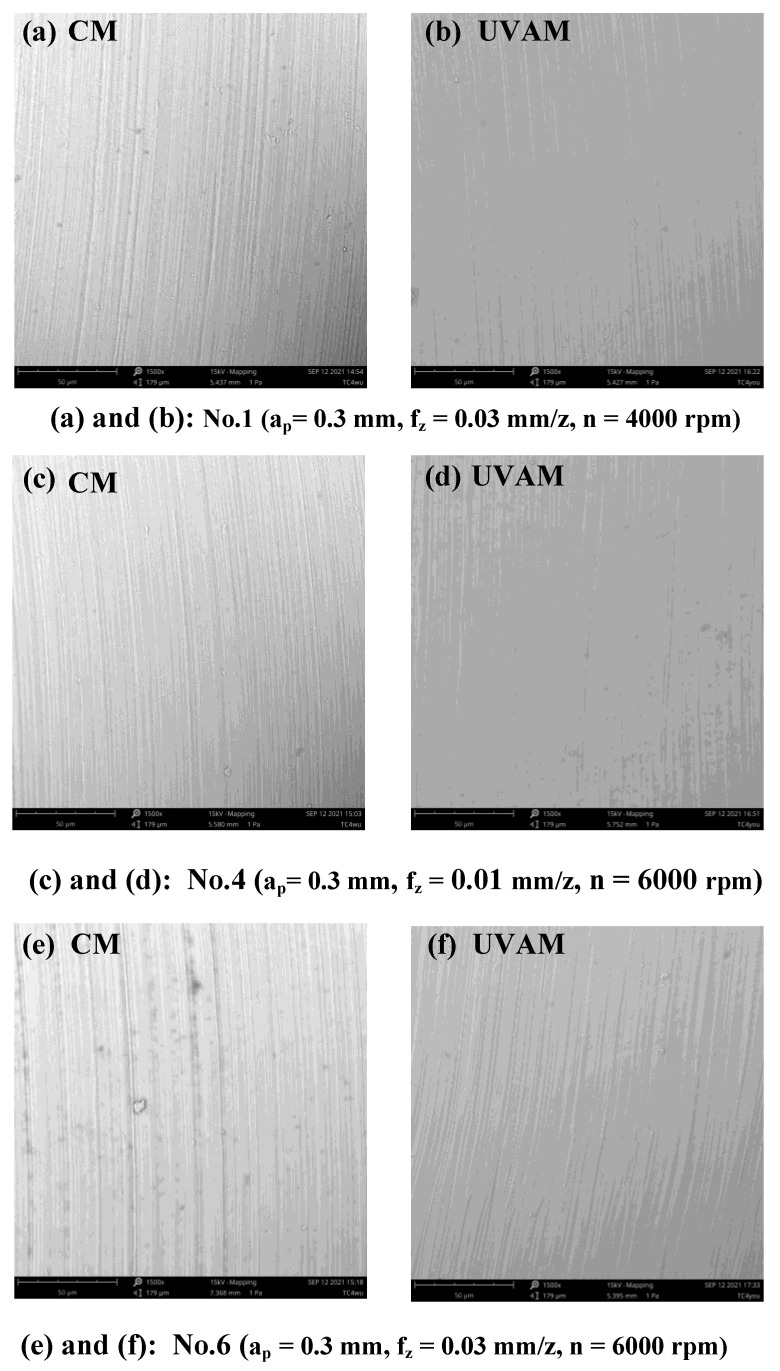
SEM topography (1500 ×) of the machined surface under CM and UVAM.

**Figure 6 micromachines-12-01319-f006:**
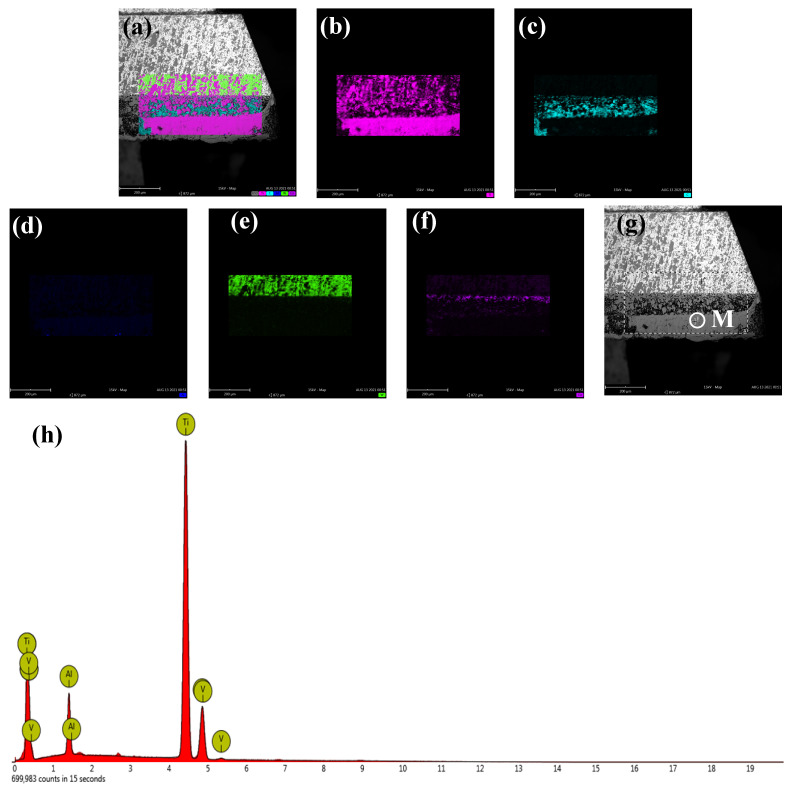
Element distribution and EDS of the flank face in ultrasonic-assisted milling: (**a**) combined map; (**b**) titanium; (**c**) carbon; (**d**) aluminum; (**e**) tungsten; (**f**) cobalt; (**g**) flank face; (**h**) EDS analysis of point M.

**Figure 7 micromachines-12-01319-f007:**
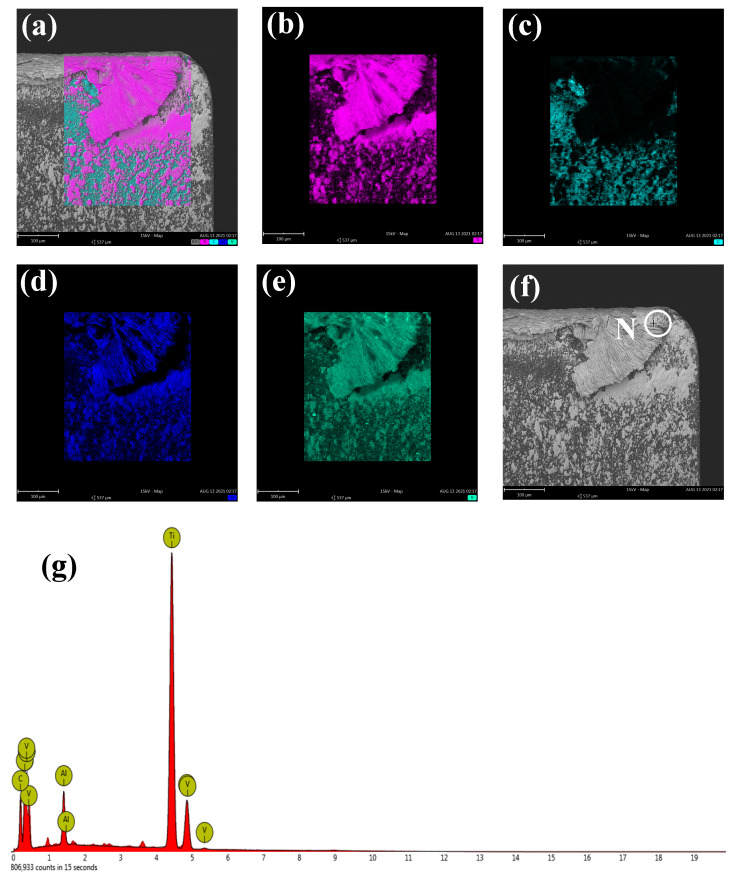
Element distribution and EDS of rake face in ultrasonic-assisted milling: (**a**) combined map; (**b**) titanium; (**c**) carbon; (**d**) aluminum; (**e**) vanadium; (**f**) rake face; (**g**) EDS analysis of point N.

**Figure 8 micromachines-12-01319-f008:**
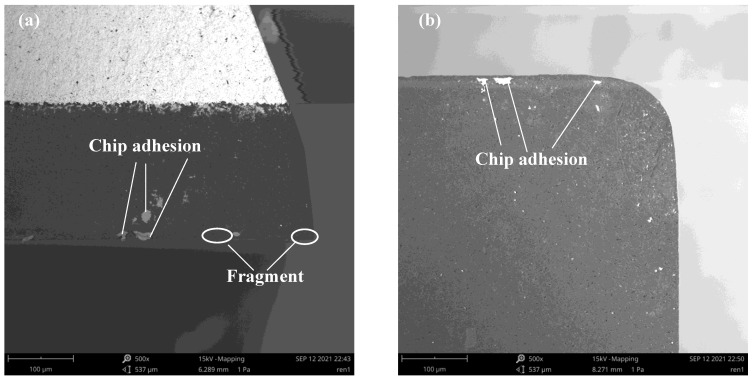
PCD tool wear of UVAM: (**a**) flank face; (**b**) rake face.

**Table 1 micromachines-12-01319-t001:** Cutting conditions.

Milling Machine	HAAS OM-2A
Workpiece	Conventional Ti6Al4V
PCD tool	Rake angle 0°, relief angle 15°
Tool diameter	4 mm
Flutes	2
Spindle speed	4000~10,000 rpm
Depth of cut a_p_	0.1~0.5 mm
Feed per tooth f_z_	0.01~0.08 mm/z
Lubrication method	Dry milling
Cutting length	612 mm
Ultrasonic vibration direction	Axial direction
Ultrasonic frequency	30 kHz
Ultrasonic amplitude	6 μm

**Table 2 micromachines-12-01319-t002:** Several groups of cutting parameters.

Number	a_p_ (mm)	f_z_ (mm/z)	n (rpm)
No. 1	0.3	0.03	4000
No. 2	0.3	0.03	8000
No. 3	0.3	0.03	10,000
No. 4	0.3	0.01	6000
No. 5	0.3	0.05	6000
No. 5	0.3	0.03	6000

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
