# Peer review of "An Investigation of Cutting Performance and Action Mechanism in Ultrasonic Vibration-Assisted Milling of Ti6Al4V Using a PCD Tool"

_micromachines, 2021, doi:10.3390/mi12111319_

Round 1
Reviewer 1 Report
- After each terminology firstly appear in this paper, it necessary to explain. For example, conventional milling,...
- Eight groups of cutting parameters were employed in this study. However, No. 2 and No. 6 are duplicate.
- Is the spindle speed in Figs. 2 and 3 6000 rpm? More detail statement is need for Figs. 2 and 3.
- From Table 2, the cutting length is 612 mm. What is the size of workpiece in this study? How the authors completed this cutting experiment?
- 6(h) and Fig. 7(g) are not clear.
- Some references in format are wrong, especially in title of Journal. Please the authors recheck it.
- Noticed some words in spelling is incorrect. For example, “…there are some main pespectives on its action mechanism” (in Sec.3.5).
- What are the number of cutting experiments and machine center in this study?
- English in the main text is recommended to be rechecked by native speakers for good understanding.
Author Response
Dear reviewers in “Micromachines ”:
I am very grateful for your comments of my manuscript. You really give us a very good chance to improve this manuscript. We have revised manuscript carefully according to your suggestions, and our response to your questions can be found as follows. In addition, all changes made to the revised manuscript have been displayed in red.
Comments and Suggestions for Authors:
Reviewer #1:
- After each terminology firstly appear in this paper, it necessary to explain. For example, conventional milling,...
The authors’ response:
Thank you for your suggestions. We have checked and revised the relevant part in the manuscript.
- Eight groups of cutting parameters were employed in this study. However, No. 2 and No. 6 are duplicate.
The authors’ response:
Thank you for your suggestions. We have revised the relevant part in section 3.2 of this manuscript.
- Is the spindle speed in Figs. 2 and 3 6000 rpm? More detail statement is need for Figs. 2 and 3.
The authors’ response:
Thank you for your suggestions. We have added the the relevant part in section 3.1 of this manuscript. The supplementary content is as follows:
In this part of the study, the cutting parameters of Fig. 2 were selected as follows: ap= 0.3 mm, fz = 0.03 mm/z ~ 0.08 mm/z, n= 6000 rpm. Similarly, the cutting parameters of Fig. 3 were selected as follows: ap= 0.1 mm ~ 0.5 mm, fz = 0.03 mm/z, n= 6000 rpm.
- From Table 2, the cutting length is 612 mm. What is the size of workpiece in this study? How the authors completed this cutting experiment?
The authors’ response:
Thank you for your question. The size of the Ti6Al4V titanium alloy is about 105mm in length, 51mm in width and 12.5 mm in height, respectively. Since this milling test is a process of linear milling grooves on the wrokpiece surface, therefore, the final cutting length is the sum of the lengths of these linear grooves.
- 6(h) and Fig. 7(g) are not clear.
The authors’ response:
Thank you for your question. We have revised the Fig. 6(h) and Fig. 7(g).
- Some references in format are wrong, especially in title of Journal. Please the authors recheck it.
The authors’ response:
Thank you for your question. We have carefully checked the references and revised the relevant format.
- Noticed some words in spelling is incorrect. For example, “…there are some main pespectives on its action mechanism” (in Sec.3.5).
The authors’ response:
Thank you for your question. We have corrected the relevant questions in the revised manuscript.
- What are the number of cutting experiments and machine center in this study?
The authors’ response:
Thank you for your question. In this study, the number of cutting experiments is 6 and the cutting experiments were carried out on a milling machine of HAAS OM-2A(Table 1 Cutting conditions)
- English in the main text is recommended to be rechecked by native speakers for good understanding.
The authors’ response:
We do agree with your advice. We have checked this paper and revised some grammatical errors in this article.

Reviewer 2 Report
The performance of ultrasonic vibration-assisted milling (UVAM) of Ti6Al4V titanium alloys was investigated with polycrystalline diamond tools, using dry milling as a benchmark. It was found that UVAM outperformed dry milling in terms of cutting force and surface quality. During ultrasonic vibration-assisted milling, PDC tools were worn due to the adhesion of workpiece materials. This manuscript demonstrates the effectiveness of the ultrasonic vibration-assisted technique in milling Ti6Al4V titanium alloys. However, there are several details that need to be improved and revised.
- Paragraphs two, three, and four of the Introduction should be placed into one paragraph since they are all relevant to ultrasonic-assisted machining.
- Please try not to use red fonts in figures.
- Testing conditions should be added in figure captions for quick reading.
- The quality of the SEM images shown in Fig. 5 needs to be improved, and the testing conditions should be added to this figure caption.
- 6(d) is unclear. ‘Fig. 6 and Fig. 7 show energy-dispersive X-ray spectroscopy (EDS) map and analysis results of the adhesion material on the flank face’. Figure 6 shows the information of the flank face. Please confirm.
- Fragments cannot be observed in the SEM image shown in Fig. 8(a). The testing conditions should be added to this caption.
- The quality of these EDS mapping images needs to be improved.
Author Response
Dear reviewers in “Micromachines ”:
I am very grateful for your comments of my manuscript. You really give us a very good chance to improve this manuscript. We have revised manuscript carefully according to your suggestions, and our response to your questions can be found as follows. In addition, all changes made to the revised manuscript have been displayed in red.
Comments and Suggestions for Authors
Reviewer #2: The performance of ultrasonic vibration-assisted milling (UVAM) of Ti6Al4V titanium alloys was investigated with polycrystalline diamond tools, using dry milling as a benchmark. It was found that UVAM outperformed dry milling in terms of cutting force and surface quality. During ultrasonic vibration-assisted milling, PDC tools were worn due to the adhesion of workpiece materials. This manuscript demonstrates the effectiveness of the ultrasonic vibration-assisted technique in milling Ti6Al4V titanium alloys. However, there are several details that need to be improved and revised.
- Paragraphs two, three, and four of the Introduction should be placed into one paragraph since they are all relevant to ultrasonic-assisted
The authors’ response:
Thank you for your suggestions. We have revised the question and paragraphs two, three and four of the Introduction have been combined in one paragraph.
- Please try not to use red fonts in figures.
The authors’ response:
Thank you for your suggestions. We have revised the red fonts of all figures in this paper.
- Testing conditions should be added in figure captions for quick reading.
The authors’ response:
Thank you for your suggestions. We have revised the relevant part in this manuscript.
- The quality of the SEM images shown in Fig. 5 needs to be improved, and the testing conditions should be added to this figure caption.
The authors’ response:
Thank you for your suggestions. We have added the corresponding cutting parameters in Fig. 5.
- 6(d) is unclear. ‘Fig. 6 and Fig. 7 show energy-dispersive X-ray spectroscopy (EDS) map and analysis results of the adhesion material on the flank face’. Figure 6 shows the information of the flank face. Please confirm.
The authors’ response:
Thank you for your suggestions. Figure 6 is not very clear because of its low aluminum content. Fig. 6 shows Element distribution and EDS of flank face in ultrasonic-assisted milling and Fig. 7 shows Element distribution and EDS of rake face in ultrasonic-assisted milling.
- Fragments cannot be observed in the SEM image shown in Fig. 8(a). The testing conditions should be added to this caption.
The authors’ response:
Thank you for your suggestions. The SEM image of elliptical area shows the fragments in following Fig. 8(a). Fig. 8(a) indicated that the image of elliptic region is not very clear and the main reason for this can attribute to the poor conductivity of PCD tool in scanning electron microscope equipment.
In this paper, all ultrasonic cutting tests were performed by one PCD tool. The PCD wear was investigated after finishing all the ultrasonic cutting tests. Therefore, all relevant cutting parameters were not listed here, and the relevant explanations of cutting parameter conditions have been added in Section 3.4.
- The quality of these EDS mapping images needs to be improved.
The authors’ response:
Thank you for your suggestions. We have revised the quality of these EDS mapping images.
Round 2
Reviewer 1 Report
No